mechanical engineering/biometrics

anisotropic superhydrophobicity, sliding angle, *Nepenthes* slippery zone, lunate cell, wax coverings, ratchet effect

**Author for correspondence:**
Lixin Wang
e-mail: wanglx@hebust.edu.cn

# Inner surface of *Nepenthes* slippery zone: ratchet effect of lunate cells causes anisotropic superhydrophobicity

Lixin Wang[1], Shuoyan Zhang[1], Shanshan Li[1], Shixing Yan[2] and Shiyun Dong[2]

[1]School of Mechanical Engineering, Hebei University of Science and Technology, Shijiazhuang 050018, People's Republic of China
[2]National Key Laboratory for Remanufacturing, Academy of Armord Forces Engineering, Beijing 100072, People's Republic of China

LW, 0000-0002-4205-5638

Inner surface of *Nepenthes* slippery zone shows anisotropic superhydrophobic wettability. Here, we investigate what factors cause the anisotropy via sliding angle measurement, morphology/structure observation and model analysis. Static contact angle of ultrapure-water droplet exhibits the value of 154.80°–156.83°, and sliding angle towards pitcher bottom and up is $2.82 \pm 0.45°$ and $5.22 \pm 0.28°$, respectively. The slippery zone under investigation is covered by plenty of lunate cells with both ends bending downward, and a dense layer of wax coverings without directional difference in morphology/structure. Results indicate that the slippery zone has a considerable anisotropy in superhydrophobic wettability that is most likely caused by the lunate cells. A model was proposed to quantitatively analyse how the structure characteristics of lunate cells affect the anisotropic superhydrophobicity, and found that the slope/precipice structure of lunate cells forms a ratchet effect to cause ultrapure-water droplet to roll towards pitcher bottom/up in different order of difficulty. Our investigation firstly reveals the mechanism of anisotropic superhydrophobic wettability of *Nepenthes* slippery zone, and inspires the bionic design of superhydrophobic surfaces with anisotropic properties.

## 1. Introduction

*Nepenthes* (Nepenthaceae) plants of the carnivorous specie have evolved a specialized organ, namely pitcher that grows from the

tip of its conspicuous leaf to efficiently capture, retain and finally digest predominantly arthropods [1–4], which obtain the nutrients required to survive in nutrient-poor habitats [5,6]. Within most of the *Nepenthes* genus, considering the considerable difference in morphology/structure and the corresponding function, the pitcher is typically distinguished as four parts: canopy-like lid, collar-formed peristome, slippery zone and digestive zone [7,8]. The slippery zone situated below the peristome is covered by plenty of lunate cells with both ends bending downward and a dense layer of wax coverings [9,10]. Most insects find the highly evolved slippery zone has considerably unique slippage properties, that execute the function of retaining prey [11]. In recent years, both the morphology/structure and the predation function of the slippery zone have gradually attracted a large number of investigations, attempting to establish bionic prototypes for developing insect slippery trapping plates [10,12–15] and other bioinspired materials with anti-adhesive properties [16–21].

Meanwhile, the slippery zone has been widely explored for its superhydrophobic wettability. From a macroscopic view, the slippery surface appears an ultra-clean morphology, and shows a static contact angle of water droplet up to 160° [22]. A further study showed that micro-scaled lunate cells and nano-scaled wax coverings together bring out the superhydrophobicity, and the latter serves a primary role [23]. Our previous study investigated the wettability of slippery zones from different *Nepenthes* species, and their static contact angles show a marginal difference (128°–156°). Theoretical analysis demonstrated that wax coverings and lunate cells are responsible for the superhydrophobicity, and their different structure parameters cause somewhat distinguishable static contact angles [24]. Lunate cells depend on their downward-oriented morphology to cause the slippery zone to exhibit a considerable anisotropy, and many studies have focused on the anisotropic properties. By behaviour observation and force measurement, authors have confirmed that the lunate cells are responsible for the anisotropic properties, as enhancing/restricting insect attachment capability in downward/upward direction [7,11,15,25,26]. Our recent study has revealed the anisotropy mechanism, as the anisotropic configuration of lunate cells produces most of the anisotropy, whereas the wax coverings generate a slight contribution [27]. These studies have explored the anisotropy effect on attachment capability and wettability behaviour, but up to now no investigation has focused on the anisotropic superhydrophobicity mechanism. In this study, we characterize the anisotropic wettability of *Nepenthes* slippery zone, and propose a model to quantitatively analyse how the structural characteristics of lunate cells affect the anisotropic superhydrophobicity. The expected results will show a comprehensive and scientific interpretation to the anisotropic superhydrophobicity of *Nepenthes* slippery zone, and inspire the bionic design of superhydrophobic surfaces with anisotropic properties.

# 2. Material and methods

## 2.1. *Nepenthes* slippery zone

Slippery zone from the mature pitcher (figure 1a, length 130.8 ± 5.5 mm, mean ± s.d. used throughout, n = 12) of *Nepenthes alata* was used to investigate the anisotropic superhydrophobicity. The *Nepenthes* plants were commercially obtained from a nursery and continuously cultivated in a greenhouse under environmental conditions: temperature 28°C–30°C and relative humidity 70%–90%. Each selected slippery zone was rinsed with distilled water to remove the contaminants (dust, pollen and wing scale). These well-prepared slippery zones were used for wettability characterization and morphology/structure observation.

## 2.2. Wettability characterization

Static contact angle and sliding angle of *Nepenthes* slippery zone were measured with an optical contact angle analyser (SL-200KS, Solon Corp., USA), to characterize the anisotropic superhydrophobicity. Several pieces (2 × 2 cm²) were cut from the well-prepared slippery zone, and horizontally fixed on a measuring platform. By a needle-in sessile drop method which has been finely described in previous literature [22,24,28,29], an ultrapure-water droplet (approx. 3 µl) was dropped on the specimen, then a high-speed imaging system and software CAST v. 3.0 belonging to the SL-200KS captured the image and calculated the static contact angle. For the sliding angle measurement, the specimen was fixed on the measuring platform, and dropped an ultrapure-water droplet (approx. 3 µl). Then the measuring platform was rotated until the droplet generated a critical roll off towards pitcher bottom/up, the software CAST 3.0 recorded the incline angle, namely the sliding angle. All these measurements were conducted under ambient conditions of temperature 28°C and relative humidity 65%.

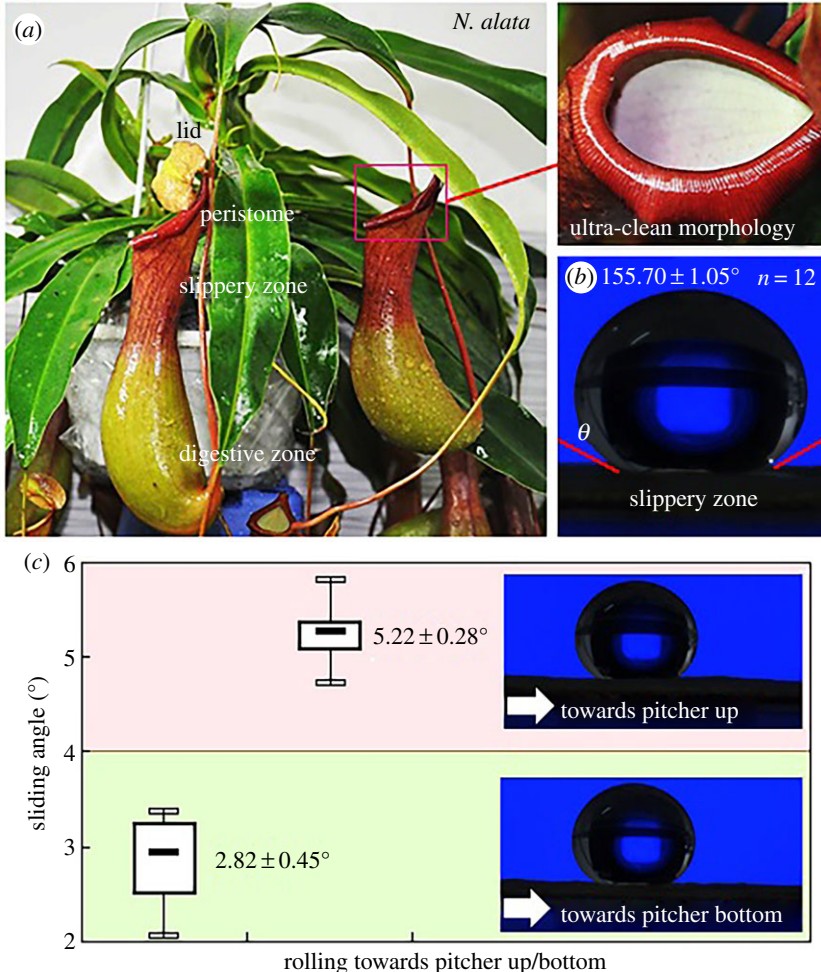

**Figure 1.** *Nepenthes* pitcher and anisotropic superhydrophobicity. (*a*) Pitcher structure and ultra-clean morphology of the slippery zone. (*b*) Nearly spherical morphology of ultrapure-water droplet on the slippery zone and its static contact angle. (*c*) Sliding angle of ultrapure-water droplet on the slippery zone rolling towards pitcher up/bottom.

## 2.3. Morphology/structure observation

To investigate how the structure characteristics of slippery zone affect the anisotropic superhydrophobicity, morphology/structure of the slippery zone was observed with a scanning electron microscope (SEM, Hitachi S-4800, Hitachi Corp., Japan) and a scanning white-light interferometer (SWLI, Zygo NV-5000, Zygo Corp., USA). Following the similar methods described in our previous investigations [8,15,24,27,30], several samples ($1 \times 1$ cm$^2$) from the well-prepared slippery zone were dried with a method of critical point drying and sputter-coated (Gatan 682 PECS, Gatan Inc.), and then examined with the SEM apparatus. Structure characteristics in a horizontal direction were estimated via analysing the SEM images. Several specimens ($2 \times 1$ cm$^2$) were cut from the well-prepared slippery zone and attached to an aluminium block, and then examined with the SWLI equipment. Structure characteristics in a vertical direction were obtained by analysing the SWLI images. All observations were conducted at an environmental temperature of 26°C and a relative humidity of 45%.

## 3. Results

### 3.1. Anisotropic superhydrophobicity

The ultrapure-water droplet on the slippery zone exhibits a nearly spherical morphology (figure 1*b*), and the static contact angle ranges from 154.80° to 156.83° with a mean value of 155.70 ± 1.05°, $n = 12$. This indicates that the *Nepenthes* slippery zone has a considerable superhydrophobicity. As shown in figure 1*c*, when the ultrapure-water droplet rolls towards pitcher bottom and up, the slippery zone

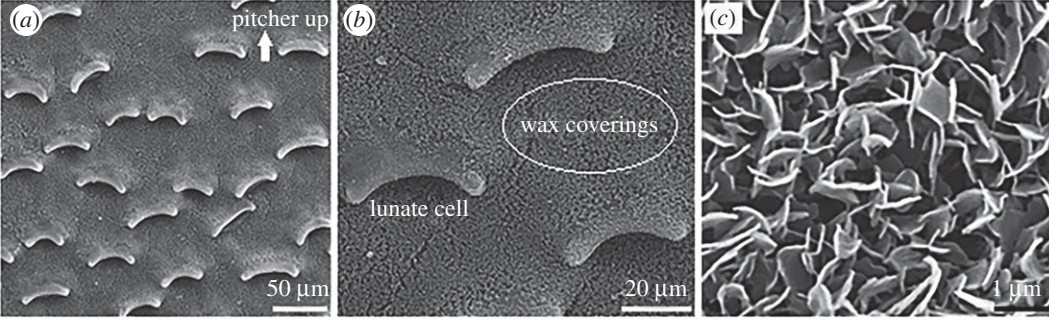

**Figure 2.** Morphology/structure of the *Nepenthes* slippery zone. (*a*) Morphology of the slippery zone. (*b*) Specific structure of the lunate cells. (*c*) Specific structure of the wax coverings, showing the platelet-shaped wax crystals and the cavities formed by overlapped wax crystals.

**Table 1.** Structure parameters of lunate cells. Number of measurement $n = 20$. Unit of the length, width, interval distance and height is µm, and unit of the angle is degrees.

| type | length | width | interval distance | slope height | precipice height | slope angle | precipice angle |
|---|---|---|---|---|---|---|---|
| values | 58.29 ± 4.51 | 14.42 ± 1.09 | 94.31 ± 8.54 | 5.39 ± 0.43 | 20.41 ± 1.73 | 23.1 ± 2.4 | 76.1 ± 4.0 |

exhibits different sliding angles, as 2.03°–3.36° and 4.72°–5.63°, respectively. The sliding angle towards pitcher up (5.22 ± 0.28°, $n = 20$) is 85.1% higher than that (2.82 ± 0.45°, $n = 20$) towards pitcher bottom. The considerable difference strongly indicates that the *Nepenthes* slippery zone has an observable anisotropic superhydrophobicity.

## 3.2. Morphology/structure

The slippery zone of *Nepenthes alata* contributes about one-third of the entire pitcher length, and its inner surface shows an ultra-clean morphology (figure 1*a*). Considerably similar to the previous investigations [8,12,14,15,27,30–32], our SEM images (figure 2*a*) show that the slippery zone consists of a dense layer of wax coverings, along with plenty of scattered lunate cells. The singly distributed lunate cell corresponds to an enlarged guard cell with both ends bending towards pitcher bottom, generating a crescent-shaped morphology with asymmetrical profile (figure 2*b*). The wax coverings are composed of discernible platelet-shaped wax crystals with irregular profile, arranging densely and mostly overlapping each other to form cavities (figure 2*c*). Viewing towards pitcher bottom and up, the lunate cells exhibit considerable difference in morphology/structure (figure 2*b*), whereas the wax coverings show highly similar arrangements and the difference in morphology/structure can hardly be detected (figure 2*c*). The lunate cells make the slippery zone bring out a considerable anisotropy in morphology/structure. By analysing the saved SEM images, the structure characteristics of lunate cells and wax coverings in a horizontal direction can be quantified. As shown in table 1, the lunate cells have approximately 10 µm scaled structure/distribution parameters, such as the length, width and interval distance, and show the distribution density of 266.7 ± 19.7 mm$^{-2}$. The wax coverings exhibit nanometre- to micrometre-ranged structure parameters, including the height (1.98 ± 0.16 µm, $n = 20$), as well as the length (1.22 ± 0.10 µm, $n = 20$) and thickness (92.5 ± 8.4 nm, $n = 20$) of the single platelet-shaped wax crystal. Exactly as the lotus effect [33,34], the micro-nano composite structure confers an extraordinary superhydrophobic wettability to the *Nepenthes* slippery zone, and it has been confirmed that the nano-scaled wax coverings serve a much more important role [23,24].

To quantify the structure characteristics of lunate cells and wax coverings in a vertical direction, SWLI observation was conducted. The slippery zone is uneven and shows a considerable variation in morphology elevation. Lunate cells protrude from the substrate of the slippery zone to present considerably different elevations towards pitcher bottom and up. Towards pitcher bottom, the lunate cell shifts slowly from underside to upside that forms a 'slope' (figure 3*a,b*). In the opposite direction, the lunate cell shifts sharply from underside to upside that forms a 'precipice' (figure 3*a,b*). The slope/

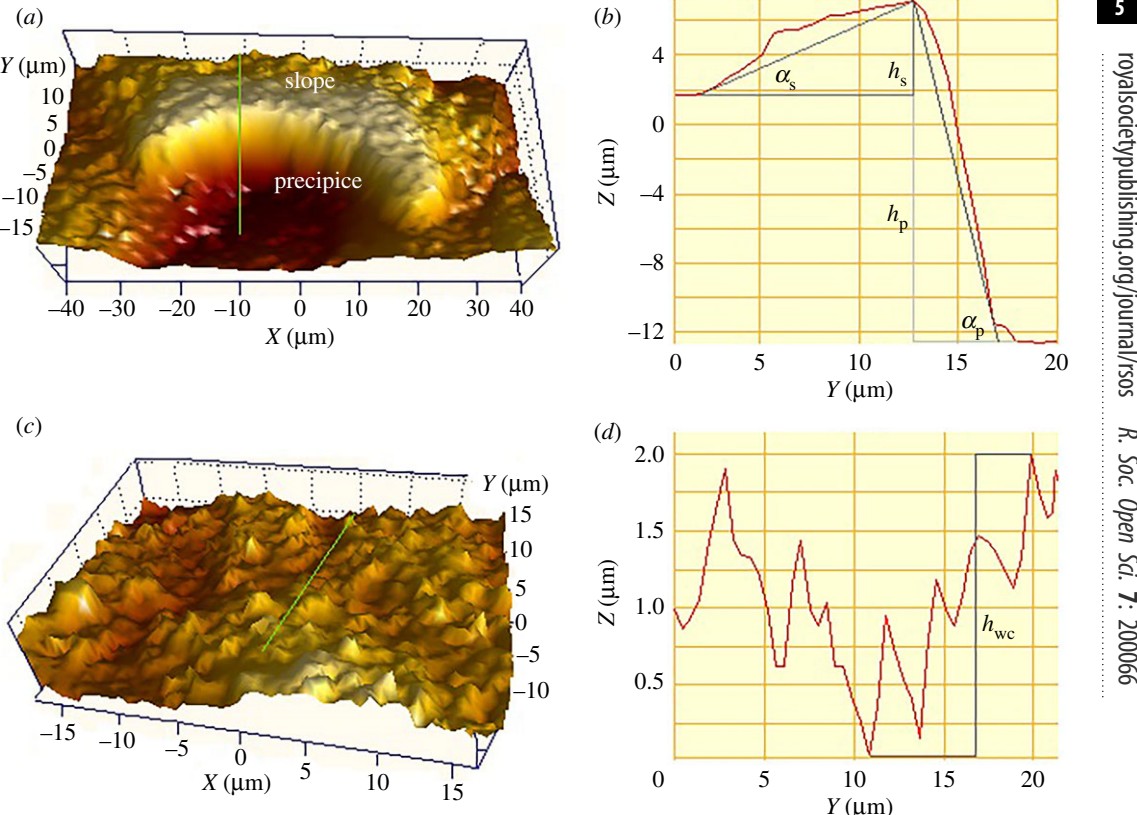

**Figure 3.** SWLI examination of the *Nepenthes* slippery zone. (*a*) SWLI image of the lunate cell. (*b*) Height variation of the lunate cell. (*c*) SWLI image of the wax coverings. (*d*) Height variation of the wax coverings. $\alpha_s/\alpha_p$, slope/precipice angle; $h_s/h_p/h_{wc}$, slope/precipice/wax coverings height.

precipice of lunate cell with remarkably different structure parameters (height, angle, table 1) highly resembles the pawl of a ratchet system that confers a considerable anisotropy to the slippery zone. Towards pitcher bottom/up, the ratchet structure would produce different effects on the sliding behaviour of a water droplet, thereby causes the slippery zone to exhibit anisotropic wettability. The wax coverings exhibit a relatively smooth morphology with a tiny elevation variation ($1.98 \pm 0.16$ µm, $n = 20$, figure 3*c*,*d*).

# 4. Discussion

Our obtained results concerning the wettability and structure characteristics indicate that the *Nepenthes* slippery zone has the anisotropic superhydrophobicity, and the ratchet effect produced by lunate cells is the exclusive causation. In the following, by model analysis, we quantitatively analyse how the lunate cells affect the anisotropic superhydrophobicity.

## 4.1. Causation of the anisotropic superhydrophobicity

According to a molecular dynamic investigation of droplet morphology on lubricant-impregnated surface [35], when an ultrapure-water droplet is sliding on the *Nepenthes* slippery zone towards pitcher up/bottom, the ultrapure-water droplet neither completely infiltrates into nor floats on the micro-nano scaled structures of the slippery zone. Instead, the ultrapure-water droplet partly infiltrates into the isotropic wax coverings and anisotropic lunate cells. Structure characteristics of the lunate cell, such as slope/precipice structure, both ends bending towards pitcher bottom and crescent-profiled morphology, confer a considerable anisotropy to the *Nepenthes* slippery zone. Plenty of convincible investigations have confirmed that the anisotropic properties produce a great effect on insect attachment capability, that effectively restricts attachment behaviour when insects are climbing on the slippery zone towards pitcher up, but obviously enhances attachment behaviour when the slippery zone is inverted [7,11,15,25,26]. A preliminary study showed the anisotropy effect

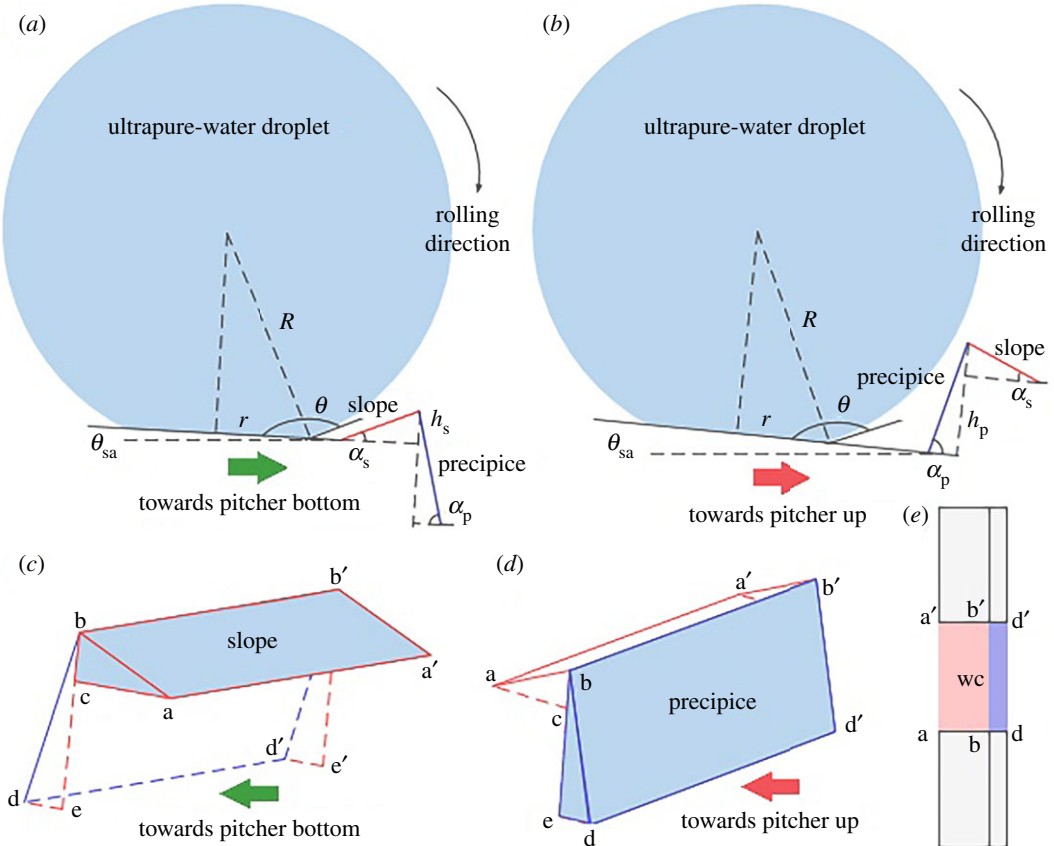

**Figure 4.** A model used to analyse the ratchet effect produced by lunate cells on the sliding angle. (*a*) Ultrapure-water droplet rolling towards pitcher bottom, wetting the slope. (*b*) Ultrapure-water droplet rolling towards pitcher up, wetting the precipice. (*c*) Ultrapure-water droplet wetting one rectangular area (aa′b′b) and two triangular areas (abc and a′b′c′) of the slope. (*d*) Ultrapure-water droplet wetting one rectangular area (bb′d′d) and two rectangular areas (bde and b′d′e′) of the precipice. (*e*) Ultrapure-water droplet wetting the wax coverings among the lunate cells, namely rectangular area aa′b′b/b b′d′d when rolling towards pitcher bottom/up.

on the wettability of slippery zone that causes the water droplet to produce a sliding angle of approximate 3° when moving towards pitcher bottom, but becomes approximate 10° when moving towards pitcher up [26]. Our recent studies explored the contribution of lunate cells and wax coverings to the anisotropic properties, and revealed that lunate cells depend on their anisotropic configuration to produce a remarkable contribution, whereas both surface topography and structure characteristics of the wax coverings produce almost no contribution [27]. Therefore, lunate cells are the exclusive causation to bring out the anisotropy of *Nepenthes* slippery zone. When measuring the sliding angle towards pitcher bottom/up, it is not the wax coverings but the lunate cells change remarkably in morphology/structure, causing the considerable difference in sliding angle. Consequently, we attempt to investigate how the structure characteristics of lunate cells affect the anisotropic superhydrophobicity of slippery zone via model analysis.

## 4.2. Mechanism analysis of anisotropic superhydrophobicity

For the lunate cell, its slope/precipice structure confers a considerable anisotropy to the *Nepenthes* slippery zone. During the sliding angle measurement, the ultrapure-water droplet is approximate 3 µl (radius 0.895 mm) and generates the static contact angle of 155.70 ± 1.05°. This indicates that the contact radius of the ultrapure-water droplet is 0.368 mm and covers about 113 lunate cells. These lunate cells with considerably anisotropic configuration and approximately 10 µm scaled structure parameters would produce an observable effect on the rolling behaviour of ultrapure-water droplet. Therefore, a model (figure 4) is proposed to quantitatively analyse the effect of lunate cells (ratchet effect) on the sliding angle.

In this model, a lunate cell is simplified as the pawl (slope/precipice structure) of a ratchet system (figure 4a,b) that has the structure parameters as shown in table 1. And, in the direction perpendicular to the ultrapure-water rolling (towards pitcher bottom/up), lunate cells are regarded as equally distributed on the slippery zone (figure 4e). Therefore, when the ultrapure-water droplet rolls, both lunate cells and wax coverings among the lunate cells are wetted. When a sliding angle is applied to the slippery zone, the ultrapure-water droplet is raised to a height and thus obtains a gravitational potential energy to achieve rolling. In another aspect, the resistance, including the gravitational potential energy produced by the slope/precipice, the adhesion/friction force between the ultrapure-water droplet and the slippery zone will restrict the rolling. When the gravitational potential energy induced by a sliding angle is larger than the work done by resistance, the ultrapure-water droplet starts to roll on the slippery zone. According to the law of energy conservation, the equations concerning the sliding angle and the structural characteristics of lunate cells can be deduced (electronic supplementary material, S1 showing the derivation), as follows:

$$mgr\sin\theta_{\text{sa-st}} = mgh_{\text{s}} + \gamma_{\text{lg}}\left[\frac{2rh_{\text{s}}}{\tan\alpha_{\text{s}}} + n\left(\frac{h_{\text{s}}}{\sin\alpha_{\text{s}}} - \frac{h_{\text{s}}}{\tan\alpha_{\text{s}}}\right)l + n\frac{h_{\text{s}}^2}{\tan\alpha_{\text{s}}}\right](1 + \sin\theta + \cos\theta)$$
$$+ \mu mg\cos\theta_{\text{sa-st}}\frac{h_{\text{s}}}{\tan\alpha_{\text{s}}}$$

and

$$mgr\sin\theta_{\text{sa-pt}} = mgh_{\text{p}} + \gamma_{\text{lg}}\left[\frac{2rh_{\text{p}}}{\tan\alpha_{\text{p}}} + n\left(\frac{h_{\text{p}}}{\sin\alpha_{\text{p}}} - \frac{h_{\text{p}}}{\tan\alpha_{\text{p}}}\right)l + n\frac{h_{\text{p}}^2}{\tan\alpha_{\text{p}}}\right](1 + \sin\theta + \cos\theta)$$
$$+ \mu mg\cos\theta_{\text{sa-pt}}\frac{h_{\text{p}}}{\tan\alpha_{\text{p}}},$$

where $\theta_{\text{sa-st}}/\theta_{\text{sa-pt}}$ is the theoretical sliding angle towards pitcher bottom/up, $mg$ is the gravity of ultrapure-water droplet, $h_{\text{s}}/h_{\text{p}}$ stands for the slope/precipice height, $\alpha_{\text{s}}/\alpha_{\text{p}}$ is the slope/precipice angle, $\theta$ represents the static contact angle, $\gamma_{\text{lg}}$ is the surface tension of ultrapure-water, $n$ stands for the number of lunate cells being wetted when the ultrapure-water droplet rolls from the underside to upside of the slope/precipice.

According to the above equations and the structure parameters (table 1), the theoretical sliding angle can be calculated, as 2.88°–3.14° towards pitcher bottom and 5.89°–8.26° towards pitcher up. The results are highly consistent with the measured values (figure 1c), indicating the proposed model can quantitatively explain the anisotropic superhydrophobicity mechanism of Nepenthes slippery zone. Further, structure parameters (angle, height) involving the ratchet effect generate a considerable influence on the sliding angle.

A negative relationship exists between the slope/precipice angle and the sliding angle. That is, with the increasing of the slope/precipice angle, the sliding angle towards pitcher bottom/up decreases considerably (figure 5a,b). When the slope/precipice height is maintained at a constant value, increasing the slope/precipice angle brings out the decrease of the slope/precipice length ($h_{\text{s}}/\tan\alpha_{\text{s}}$, $h_{\text{p}}/\tan\alpha_{\text{p}}$, figure 4a,b), thereby the work done by the adhesion/friction force also decreases accordingly. In this situation, a smaller sliding angle can generate the gravitational potential energy that causes the ultrapure-water droplet to roll. On the contrary, the sliding angle increases/decreases significantly when the slope/precipice height is increased/decreased, existing as a positive relationship (figure 5c,d). Increasing/decreasing the slope/precipice height will produce a positive effect on the gravitational potential energy which restricts the rolling of ultrapure-water droplet. The slope/precipice length ($h_{\text{s}}/\tan\alpha_{\text{s}}$, $h_{\text{p}}/\tan\alpha_{\text{p}}$, figure 4a,b) shows the same trend with the slope/precipice height, thus increasing/decreasing the slope/precipice height equals to increasing/decreasing the work done by the adhesion/friction force. Therefore, increasing/decreasing the slope/precipice height needs a larger/smaller sliding angle to cause the rolling of ultrapure-water droplet.

Previous studies have shown some extremely important theoretical models to characterize the super-slippage properties of the Nepenthes slippery zone, in the aspects of superhydrophobicity [17,26,36], insect attachment ability [15,26,27] and static contact angle [24]. Here, our proposed model firstly explains how the structural characteristics of lunate cells, including the slope/precipice angle and the slope/precipice height, affect the anisotropic superhydrophobic wettability of Nepenthes slippery zone. Results calculated from the model exhibit a clear relationship between the structural characteristics of lunate cells and the sliding angle of Nepenthes slippery zone, which can be used to manipulate the

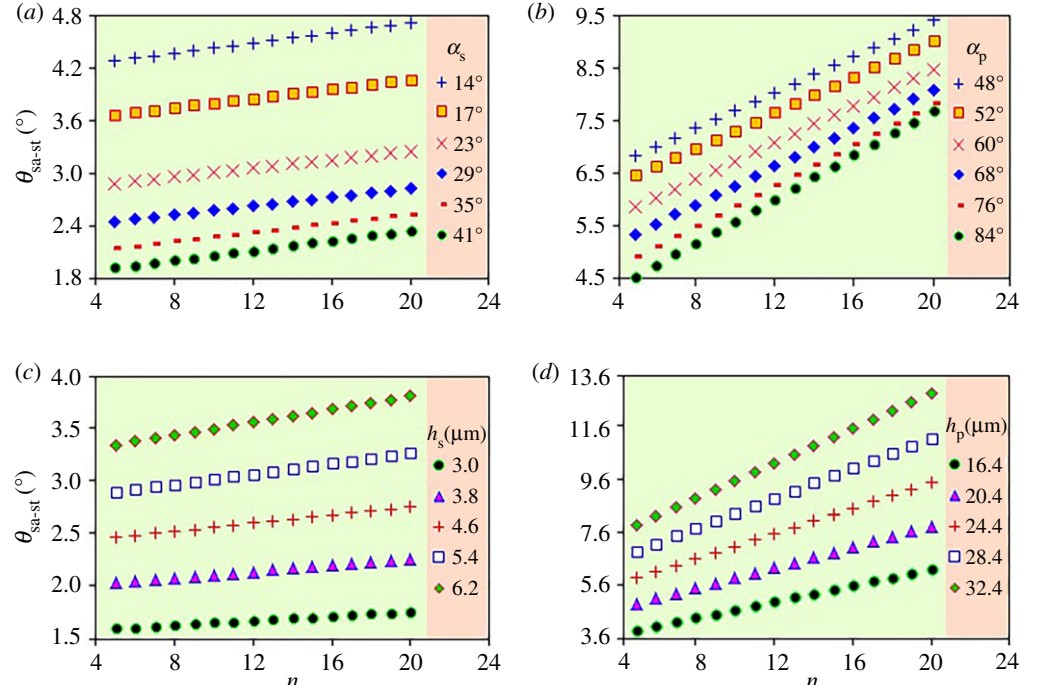

**Figure 5.** Calculation of the proposed model. (*a*) Effect of the $\alpha_s$ on the $\theta_{sa\text{-}st}$. (*b*) Effect of the $\alpha_p$ on the $\theta_{sa\text{-}pt}$. (*c*) Effect of the $h_s$ on the $\theta_{sa\text{-}st}$. (*d*) Effect of the $h_p$ on the $\theta_{sa\text{-}pt}$.

values of sliding angle towards pitcher bottom/up. More importantly, this model provides a specific principle for the bionic design of anisotropic hydrophobic surfaces with controllable effect.

# 5. Conclusion

In summary, anisotropic wettability of the *Nepenthes* slippery zone was comprehensively characterized via sliding angle measurement, morphology/structure observation. The static contact angle of ultrapure-water droplet on the slippery zone ranges from 154.80° to 156.83°, and the sliding angle towards pitcher bottom (2.82 ± 0.45°) is considerably smaller than that towards pitcher up (5.22 ± 0.28°), indicating the slippery zone has an observable anisotropy in superhydrophobic wettability. A model was proposed to quantitatively explain how the structure characteristics of lunate cells affect the anisotropic superhydrophobicity. By equations derived from the model, the quantitative relationship between the structure characteristics of lunate cells and the sliding angle has been established. The calculated sliding angle shows the value of 2.88°–3.14° towards pitcher bottom and 5.89°–8.26° towards pitcher up, which highly resembles the measured values. This study firstly reveals the anisotropic superhydrophobic mechanism of *Nepenthes* slippery zone, and is helpful for inspiring the bionic design of anisotropic superhydrophobic surfaces.

Ethics. Specimens used in this study only include *Nepenthes alata*, no special collecting permit or 'Animal Care Protocol' was required. No fieldwork was required for this study.

Data accessibility. All original data supporting this article have been uploaded as part of the electronic supplementary material.

Authors' contributions. All authors contributed to the design of the experimental approach and the model analysis. L.W. and S.D. prepared all the samples for experiment. L.W., S.L. and S.Y. interpreted the results, collected and analysed the data. L.W. and S.Z. and wrote the manuscript. All authors gave final approval for publication and agreed to be held accountable for the work performed therein.

Competing interests. We declare we have no competing interest.

Funding. This work was supported by the National Natural Science Foundation of China (grant no. 51205107), the Top Talents Program of Hebei Higher Education (grant no. BJ2017011), the Natural Science Foundation of Hebei province (grant no. E2019208306) and the Tribology Science Fund of State Key Laboratory of Tribology (grant no. SKLTKF16B01).

Acknowledgements. We acknowledge Lei Zhao (TsingHua University) for the help in performing the scanning white-light interferometer examination.

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
