## [Reviewer comments · Royal Society Open Science]

Review History

RSOS-200066.R0 (Original submission)

Review form: Reviewer 1

Is the manuscript scientifically sound in its present form?

Yes

Are the interpretations and conclusions justified by the results?

Yes

Is the language acceptable?

Yes

Do you have any ethical concerns with this paper?

No

Have you any concerns about statistical analyses in this paper?

No

Recommendation?

Accept with minor revision (please list in comments)

Comments to the Author(s)

This paper well demonstrated the anisotropic superhydrophobic of the pitcher plant's slippery zone. The novel anisotropic phenomenon, the clear structure analyzing and the reasonable model building about the ratchet effect of the water rolling on the lunate cells are reported with the essential experiment and the logical calculations. This article fits well with the journal: Royal Society Open Science and put forward a completely new point on the Nepenthes research and the design of anisotropic superhydrophobic surfaces. The submission can be published after minor revision.

1. In the results part, only used the droplet of volume $3 \mu\text{l}$ to test the sliding angle. For the accuracy of the experiment, the author was advised to do more experiment with different volume.
2. It is generally believed that the pitcher plant's surface is a kind of super-slippery surface, the author is recommended to compare the model's difference between traditional super-slippery surface with the established model in this paper.
3. In the section 1 on the introduction of the Nepenthes surface with anisotropic wettability, the authors left out a few related works such as Adv Mater 2014, 26 (19), 313; Adv Funct Mater 2019, 29, 1904446, etc.

Review form: Reviewer 2

Is the manuscript scientifically sound in its present form?

Yes

Are the interpretations and conclusions justified by the results?

Yes

Is the language acceptable?

Yes

Do you have any ethical concerns with this paper?

No

Have you any concerns about statistical analyses in this paper?

No

Recommendation?

Accept with minor revision (please list in comments)

Comments to the Author(s)

This manuscript experimentally studied the anisotropic wettability of the Nepenthes slippery zone. It was found that the anisotropic superhydrophobicity was affected by structure characteristics of lunate cells, which was also verified by the derived quantitative model. This work is expected to offer inspirations for bionic design of anisotropic superhydrophobic surfaces. Yet, I have some concerns below:

1. The sliding angle towards pitcher up is around 5.22° while the sliding angle towards pitcher bottom is around 2.82° . Thus, a conclusion that the Nepenthes slippery zone has a remarkable anisotropic superhydrophobicity is claimed. My argument is, such a difference of the sliding angle is large enough to support the conclusion?

2. This manuscript concentrate on the anisotropic wettability of *Nepenthes* slippery zone. Thus, in addition to directions of pitcher up and pitcher bottom, sliding angles of other two directions need to be measured and studied as well.
3. Chen et al. (Chen, Zhang, Zhang, Liu, Jiang, Zhang, Hang, Jiang. Continuous directional water transport on the peristome surface of *Nepenthes alata*. *Nat.* 532, 85-89.) studied the morphology of the carnivorous plant *Nepenthes alata*. Different from the sliding droplets observed in the present experiment, Chen et al. found that water would spread directionally on *Nepenthes alata*. What is the reason causing the different droplet moving behaviors?
4. From the rough morphology structure of the *Nepenthes* slippery zone as well as the excellent superhydrophobicity, it seems that the droplet on the slippery zone falls in the state I in the Molecular Dynamics study of Guo et al. (Guo, Tang, Kumar. Droplet morphology and mobility on lubricant-impregnated surfaces: a molecular dynamic study. *Langmuir*, 35, 16377-16387, 2019). More discussions of droplet morphology would deepen this work by comparing with the existing theoretical work.

Decision letter (RSOS-200066.R0)

06-Feb-2020

Dear Dr Wang

On behalf of the Editors, I am pleased to inform you that your Manuscript RSOS-200066 entitled "Inner surface of *Nepenthes* slippery zone: Ratchet effect of lunate cells causes anisotropic superhydrophobicity" has been accepted for publication in Royal Society Open Science subject to minor revision in accordance with the referee suggestions. Please find the referees' comments at the end of this email.

The reviewers and handling editors have recommended publication, but also suggest some minor revisions to your manuscript. Therefore, I invite you to respond to the comments and revise your manuscript.

- Ethics statement

- Data accessibility

If you wish to submit your supporting data or code to Dryad (<http://datadryad.org/>), or modify your current submission to dryad, please use the following link:
<http://datadryad.org/submit?journalID=RSOS&manu=RSOS-200066>

- **Competing interests**

- **Authors' contributions**

- **Acknowledgements**

- **Funding statement**

Because the schedule for publication is very tight, it is a condition of publication that you submit the revised version of your manuscript before 15-Feb-2020. Please note that the revision deadline will expire at 00.00am on this date. If you do not think you will be able to meet this date please let me know immediately.

If your manuscript is newly submitted and subsequently accepted for publication, you will be asked to pay the article processing charge, unless you request a waiver and this is approved by Royal Society Publishing. You can find out more about the charges at <https://royalsocietypublishing.org/rsos/charges>. Should you have any queries, please contact openscience@royalsociety.org.

on behalf of Dr Derek Abbott (Associate Editor) and R. Kerry Rowe (Subject Editor)
openscience@royalsociety.org

Reviewer comments to Author:

Reviewer: 1

Comments to the Author(s)

This paper well demonstrated the anisotropic superhydrophobic of the pitcher plant's slippery zone. The novel anisotropic phenomenon, the clear structure analyzing and the reasonable model

building about the ratchet effect of the water rolling on the lunate cells are reported with the essential experiment and the logical calculations. This article fits well with the journal: Royal Society Open Science and put forward a completely new point on the *Nepenthes* research and the design of anisotropic superhydrophobic surfaces. The submission can be published after minor revision.

1. In the results part, only used the droplet of volume $3 \mu\text{l}$ to test the sliding angle. For the accuracy of the experiment, the author was advised to do more experiment with different volume.
2. It is generally believed that the pitcher plant's surface is a kind of super-slippy surface, the author is recommended to compare the model's difference between traditional super-slippy surface with the established model in this paper.
3. In the section 1 on the introduction of the *Nepenthes* surface with anisotropic wettability, the authors left out a few related works such as *Adv Mater* 2014, 26 (19), 313; *Adv Funct Mater* 2019, 29, 1904446, etc.

Reviewer: 2

Comments to the Author(s)

This manuscript experimentally studied the anisotropic wettability of the *Nepenthes* slippery zone. It was found that the anisotropic superhydrophobicity was affected by structure characteristics of lunate cells, which was also verified by the derived quantitative model. This work is expected to offer inspirations for bionic design of anisotropic superhydrophobic surfaces. Yet, I have some concerns below:

1. The sliding angle towards pitcher up is around 5.22° while the sliding angle towards pitcher bottom is around 2.82° . Thus, a conclusion that the *Nepenthes* slippery zone has a remarkable anisotropic superhydrophobicity is claimed. My argument is, such a difference of the sliding angle is large enough to support the conclusion?
2. This manuscript concentrate on the anisotropic wettability of *Nepenthes* slippery zone. Thus, in addition to directions of pitcher up and pitcher bottom, sliding angles of other two directions need to be measured and studied as well.
3. Chen et al. (Chen, Zhang, Zhang, Liu, Jiang, Zhang, Hang, Jiang. Continuous directional water transport on the peristome surface of *Nepenthes alata*. *Nat.* 532, 85-89.) studied the morphology of the carnivorous plant *Nepenthes alata*. Different from the sliding droplets observed in the present experiment, Chen et al. found that water would spread directionally on *Nepenthes alata*. What is the reason causing the different droplet moving behaviors?
4. From the rough morphology structure of the *Nepenthes* slippery zone as well as the excellent superhydrophobicity, it seems that the droplet on the slippery zone falls in the state I in the Molecular Dynamics study of Guo et al. (Guo, Tang, Kumar. Droplet morphology and mobility on lubricant-impregnated surfaces: a molecular dynamic study. *Langmuir*, 35, 16377-16387, 2019). More discussions of droplet morphology would deepen this work by comparing with the existing theoretical work.

Author's Response to Decision Letter for (RSOS-200066.R0)

See Appendix A.

RSOS-200066.R1 (Revision)

Review form: Reviewer 1

Is the manuscript scientifically sound in its present form?

Yes

Are the interpretations and conclusions justified by the results?

Yes

Is the language acceptable?

Yes

Do you have any ethical concerns with this paper?

No

Have you any concerns about statistical analyses in this paper?

No

Recommendation?

Accept as is

Comments to the Author(s)

The authors have addressed my concerns. I'd like to recommend its acceptance in current version.

Review form: Reviewer 2

Is the manuscript scientifically sound in its present form?

Yes

Are the interpretations and conclusions justified by the results?

Yes

Is the language acceptable?

Yes

Do you have any ethical concerns with this paper?

No

Have you any concerns about statistical analyses in this paper?

No

Recommendation?

Accept as is

Comments to the Author(s)

I recommend acceptance in the present form.

Decision letter (RSOS-200066.R1)

26-Feb-2020

Dear Dr Wang,

It is a pleasure to accept your manuscript entitled "Inner surface of Nepenthes slippery zone: Ratchet effect of lunate cells causes anisotropic superhydrophobicity" in its current form for publication in Royal Society Open Science. The comments of the reviewer(s) who reviewed your manuscript are included at the foot of this letter.

on behalf of Dr Derek Abbott (Associate Editor) and R. Kerry Rowe (Subject Editor)
openscience@royalsociety.org

Reviewer comments to Author:
Reviewer: 1

Comments to the Author(s)
The authors have addressed my concerns. I'd like to recommend its acceptance in current version.

Reviewer: 2

Comments to the Author(s)
I recommend acceptance in the present form.

Appendix A

Dear editors,

Thanks for your effort to this manuscript, also appreciate the reviewers for their valuable and constructive comments. In the following, we try our best to reply the comments in detail.

Reviewer 1-Comment 1: In the results part, only used the droplet of volume 3 μ l to test the sliding angle. For the accuracy of the experiment, the author was advised to do more experiment with different volume.

Reply: Thanks for the suggestion. In fact, in measuring all the sliding angles, considering the evaporation and the syringe accuracy, the volume of ultrapure-water droplet is not strictly limited to 3 μ l, sometimes more than 3 μ l, sometimes less than 3 μ l. In order to express this point accurately, in the revised manuscript, we put a word 'approximate' before 3 μ l and have highlighted this changed point with red colour.

Reviewer 1-Comment 2: It is generally believed that the pitcher plant's surface is a kind of super-slippy surface, the author is recommended to compare the model's difference between traditional super-slippy surface with the established model in this paper.

Reply: Exactly, the *Nepenthes* pitcher consists of lid, peristome, slippy zone and digestive zone. The well-known 'super-slippy surface' is the inner surface of slippy zone. It shows super-slippage properties to insects (Roy. Soc. Open Sci. 2018, 5, 180766.). The super-slippy surface causes the water-droplet to present a sliding angle of about 3° when sliding toward pitcher bottom, and about 6° when sliding toward pitcher up. In the manuscript, our established model gives an explanation to the difference in the two types of sliding angles. Compared with the previous theories, our model is the first to focus on the difference in sliding angles.

According to the reviewer's suggestion, in the revised manuscript, we analyze the model's difference between the traditional super-slippy surface and our established model, as follows:

Previous studies have shown some extremely important theoretical models to characterize the super-slippage properties of the *Nepenthes* slippy zone, in the aspects of superhydrophobicity [17,24,33], insect attachment ability [15,24,25], and static contact angle [22]. Here, our proposed model firstly explains how the structure characteristics of lunate cells, including the slope/precipice angle and the slope/precipice height, affect the anisotropic superhydrophobic wettability of *Nepenthes* slippy zone.

The changed point has been highlighted with blue colour in the revised manuscript.

Reviewer 1-Comment 3: In the section 1 on the introduction of the *Nepenthes* surface with anisotropic

wettability, the authors left out a few related works such as *Adv Mater* 2014, 26 (19), 313; *Adv Funct Mater* 2019, 29, 1904446, etc.

Reply: Thanks for the valuable suggestion. We have read the articles suggested by the Reviewer, and have added in the revised manuscript, as Refs [20] and [21]. And, the latter references are renumbered.

20. Li P, Cao MY, Bai HY, Zhao TH, Ning YZ, Wang XS, Liu KS, Jiang L. 2019 Unidirectional liquid manipulation *via* an integrated mesh with orthogonal anisotropic slippery tracks. *Adv. Funct. Mater.* **29**, 1904446. (doi:10.1002/adfm.201904446)

21. Zhang PC, Liu HL, Meng JX, Yang G, Liu XL, Wang ST, Jiang L. 2014 Grooved organogel surfaces towards anisotropic sliding of water droplets. *Adv. Mater.* **26**, 3131-3135. (doi: 10.1002/adma.201305914)

All the changed points have been highlighted with **green color** in the revised manuscript.

Reviewer 2-Comment 1: 1. The sliding angle towards pitcher up is around 5.22 ° while the sliding angle towards pitcher bottom is around 2.82 °. Thus, a conclusion that the *Nepenthes* slippery zone has a **remarkable** anisotropic superhydrophobicity is claimed. My argument is, such a difference of the sliding angle is large enough to support the conclusion?

Reply: Thanks for pointing the inappropriate aspect. Indeed, the word ‘remarkable’ is not suitable for describing the difference between 5.22° and 2.82°. We have changed the word ‘remarkable’ with ‘observable’ or ‘obvious’ in the revised manuscript, thank you. And, the changed points are highlighted with **dark-red color**.

Reviewer 2-Comment 2: This manuscript concentrates on the anisotropic wettability of *Nepenthes* slippery zone. Thus, in addition to directions of pitcher up and pitcher bottom, sliding angles of other two directions need to be measured and studied as well.

Reply: As shown in Figure 2b and Figure 2a, based on the structure characteristics of lunate cells in the direction towards pitcher up (forming ‘precipice’) and bottom (forming ‘slope’), sliding angles of the two types can almost adequately characterize the anisotropy of *Nepenthes* slippery zone. Further, in previous studies, many authors have used the two directions to characterize the anisotropy of *Nepenthes* slippery zone in insect attachment behaviors (*Beilstein J. Nanotechnol.* **2**, 302–310. *J. Bionic Eng.* **12**, 79–87. *J. Bionic Eng.* **13**, 373–387. *Roy. Soc. Open Sci.* **5**, 180766.).

Reviewer 3-Comment 1: Chen et al. (Chen, Zhang, Zhang, Liu, Jiang, Zhang, Hang, Jiang. Continuous directional water transport on the peristome surface of *Nepenthes alata*. *Nature* 532, 85-89.) studied the morphology of the carnivorous plant *Nepenthes alata*. Different from the sliding droplets observed in the present experiment, Chen et al. found that water would spread directionally on *Nepenthes alata*. What is the reason causing the different droplet moving behaviors?

Reply: The *Nepenthes* pitcher consists of four parts: lid, peristome, slippery zone and digestive zone. In Chen's article (*Nature*, 2016, 532, 85-89), the authors focused on the peristome, they studied the droplet moving behaviors on the peristome (Figure R1). In our manuscript, we investigated the sliding behaviors of water droplet on the slippery zone (Figure R1).

Figure R1 Peristome and slippery zone of the *Nepenthes* pitchers

Reviewer 2-Comment 4: From the rough morphology structure of the *Nepenthes* slippery zone as well as the excellent superhydrophobicity, it seems that **the droplet on the slippery zone the state I in the Molecular Dynamics study** of Guo et al. (Guo, Tang, Kumar. Droplet morphology and mobility on lubricant-impregnated surfaces: a molecular dynamic study. *Langmuir*, 35, 16377-16387, 2019). More discussions of droplet morphology would deepen this work by comparing with the existing theoretical work.

Reply: Thanks for the valuable suggestion, and we have read this article. According to this article, we discussed the ultra-water droplet morphology on the slippery zone, as the follows:

According to a molecular dynamic investigation of droplet morphology on lubricant-impregnated surface [35], when an ultrapure-water droplet is sliding on the *Nepenthes* slippery zone towards pitcher up/bottom, the ultrapure-water droplet neither completely infiltrates into nor floats on the micro-nano scaled structures of slippery zone. Instead, the ultrapure-water droplet partly infiltrates into the isotropic wax coverings and anisotropic lunate cells.

And, this changed point has been highlighted with orange color in the revised manuscript.

The above are our reply to the comments. However, we are not sure that the reviewers can completely accept our reply. If not, we will revise this manuscript again, thank you.

Best regards,

Lixin Wang